# Ion-Specific Gelation and Internal Dynamics of Nanocellulose Biocompatible Hybrid Hydrogels: Insights from Fluctuation Analysis

**DOI:** 10.3390/gels11030197

**Published:** 2025-03-12

**Authors:** Arianna Bartolomei, Elvira D’Amato, Marina Scarpa, Greta Bergamaschi, Alessandro Gori, Paolo Bettotti

**Affiliations:** 1Nanoscience Laboratory, Department of Physics, University of Trento, v. Sommarive 14, Povo, 38123 Trento, Italy; aribart@ugr.es (A.B.); marina.scarpa@unitn.it (M.S.); 2National Research Council of Italy, Istituto di Chimica del Riconoscimento Molecolare (ICRM), Via Mario Bianco, 9, 20131 Milano, Italy; greta.bergamaschi@cnr.it (G.B.); alessandro.gori@cn.it (A.G.)

**Keywords:** nanocellulose, hydrogels, dynamic light scattering, sol–gel transition, cell culture

## Abstract

Hydrogels find widespread use in bioapplications for their ability to retain large amounts of water while maintaining structural integrity. In this article, we investigate hybrid hydrogels made of nanocellulose and either amino–polyethylenglycol or sodium alginates and we present two novel results: (1) the biocompatibility of the amino-containing hybrid gel synthesized using a simplified receipt does not require any intermediate synthetic step to functionalize either component and (2) the fluctuation in the second-order correlation function of a dynamic light scattering experiment provides relevant information about the characteristic internal dynamics of the materials across the entire sol–gel transition as well as quantitative information about the ion-specific gel formation. This novel approach offers significantly better temporal (tens of μs) and spatial (tens of μm) resolution than many other state-of-the-art techniques commonly used for such analyses (such as rheometry, SAXS, and NMR) and it might find widespread application in the characterization of nano- to microscale dynamics in soft materials.

## 1. Introduction

Hydrogels are defined as three-dimensional networks capable of absorbing a large amount of liquid. Given such a broad definition, their composition and physico-chemical properties are highly tunable, as are their preparation and characterization methods [1]. So, hydrogels are considered versatile materials with a wide range of applications [2]. For instance, by using the appropriate components, smart functional hydrogels can respond to various external stimuli (such as pH, temperature, and electric and mechanical stimuli) [3,4] or enable applications in advanced electronics [5], the food industry [6], and biomedicine [7,8,9]. The expected increase in large-scale applications requires moving toward sustainability and the use of natural polymers as hydrogel components, which offers an eco-friendly opportunity. In this regard, nanomaterials derived from cellulose sources are ideal candidates [10,11,12,13] with cellulose nanocrystals (CNCs) showing polymer-like behavior, which quickly jellify upon the addition of multivalent cations. The ionotropic gelation process is mediated by the interaction of the cations with the negatively charged or polar groups of CNC (mainly hydroxyls, carboxyls, carbonyls, and sulfates, depending on the CNC’s synthesis procedure) and allows precise control over the macroscopic gel structure [14,15,16] and the synthesis of hydrogels with controlled properties [17,18,19,20,21]. However, to overcome intrinsic CNC stiffness and obtain desired mechanical properties, CNCs are often used as fillers for softer polymers [22,23,24,25,26,27]. The development of nanocellulose hybrid gels as substrates for cell cultures requires both an optimized synthesis process and a thorough characterization of their physical properties.

In this study, we present two novel results: (1) we use a simplified synthesis approach that enhances the reproducibility and scalability of the hybrid material while maintaining its structural integrity and biocompatibility and, (2) to further investigate the gel’s physicochemical properties, we introduce a novel analysis of the correlation function obtained from dynamic light scattering (DLS) measurements. This refined analysis provides deeper insight into the gel network dynamics, shedding light on particle interactions and aggregation behavior.

By linking the synthesis process with advanced characterization, we establish a more comprehensive understanding of how material properties influence the performance of cell cultures, paving the way for improved biomaterials in tissue engineering and regenerative medicine applications.

Simplified synthesis to fabricate a hybrid gel made of CNC and aminated polyethylene glycol (PEG) is an important result since this formulation strategically integrates materials with complementary chemical and mechanical properties, and it enables the creation of versatile hydrogel platforms. In fact, while cellulose contributes with carboxyl groups, amino–PEG introduces valuable amine functionalities to the matrix, establishing a dual-functionality system. The presence of both amine and carboxyl groups is particularly advantageous for bio-applications, as these functional groups can facilitate cell adhesion, protein binding, and potential crosslinking through various chemical reactions such as amide bond formation. Amine functionalities can be directly added to CNC [28,29] but the reaction is time-consuming and care should be taken to avoid unexpected gelation and the cross-linking of amine and carboxylic groups. Thus, this strategy is mainly limited to dilute nanocellulosic solutions. Several studies report on the use of amino–PEG and CNC to form hybrid gels, but they mostly use diacrylate–PEG and require a dedicated step to cross-polymerize PEG and CNC. For example, in [30,31], the authors report the synthesis of a photopolymerizable 3D PEG diacrylate (PEGDA) ink in which CNC is used as a filler to increase the mechanical properties. Similarly, refs. [32,33] report the use of PEGDA-CNC composites to form gels with variable Poisson ratios. In [34], a 3D-printable bioink is formed by mixing a nanocellulosic solution with a four-arm start PEG polymer and by photopolymerizing the resulting viscous fluid. In [35], the synthesis of a hybrid hydrogel is achieved by crosslinking PEG and CNC with acrylamide and N,$N^{\prime }$-methylene bis-acrylamide. In [36], an eight-arm, branched amino–PEG polymer is acrylated and photopolymerized under UV irradiation. Ref. [37] is the only report that demonstrates the possibility of obtaining hydrogels by directly mixing micron-sized cellulosic fibers with branched, tetra-arm amino–PEG. Our approach takes a step forward compared to what has been shown so far and demonstrates the possibility of synthesizing biocompatible hybrid gels by simply mixing CNCs and linear amino–PEG. Moreover, although the biocompatibility of hybrid gels made of CNCs and PEG has been demonstrated [34,38,39], similar results are unreported for amino–PEG and CNC hybrid gels.

From the perspective of characterization methods, we develop an innovative approach to analyze DLS measurements that exploit the statistical properties of the second-order correlation function measured from commercial DLS instruments with a single scattering angle to obtain information about the sol–gel transition and material composition. To demonstrate this possibility, we exploit the peculiar characteristic of alginates to perform jellification in the presence of ${\mathrm{Ca}}^{2+}$ but not with ${\mathrm{Mg}}^{2+}$ [40,41] and we show that the statistical properties of the measured second-order correlation function contain information about the sol–gel transition dynamics and also about the material composition. We define a phenomenological Figure of Merit (FOM), based on the short-time values of the second-order correlation function, and the comparison among the two families of materials (i.e., the ${\mathrm{Ca}}^{2+}$ crosslinked gel and the ${\mathrm{Mg}}^{2+}$ suspension) unveils the differences in their internal dynamics.

The cation complexes of alginates have an intriguing nature and their role in hydrogel network formation is currently a subject of research [42]. Different methods spanning from the atomic scale up to the microscale are necessary to characterize the structural morphology, composition, and mechanical properties of hydrogels. The advantages and disadvantages of these methods have been recently reviewed and it has been concluded that hydrogel characterization remains challenging also because of the hierarchical length scales of the network structure [43]. In general, the most popular approach to investigate the macroscopic mechanical properties is rheological analysis. This method allows for the investigation of the sol–gel transition and provides quantitative information about the macroscopic properties of the material [44]. Yet, routine rheology experiments cannot sense the fine details of the gel structures and different materials can share similar macroscopic mechanical properties (like the elastic and viscous moduli) while differing in their local structure. Advanced rheological techniques such as dynamic shear rheology and large-amplitude oscillatory shear experiments are very promising investigative tools for the dynamics of the gelation of thixotropic suspensions. However, the complexity of data analysis and the necessity of integrative structural information restrict their use to few materials and require skilled researchers [45].

Dynamic light scattering (DLS) is a family of techniques that exploit the scattering of light to investigate structural and dynamical details of colloidal particles and gel dynamics. Although their basic theory has been thoroughly developed, starting from the 1950s [46,47,48,49], the complex interpretation of data and the strong assumptions on which the models are based make DLS a lively research field even today, with several variants proposed in recent years to overcome some of the limits of the techniques [50,51,52]. One of the weak points of experimental data interpretation in any DLS technique is the assumption about the type of motion of the particles and/or their size distribution. In fact, these methods require a transformation from the experimentally measured quantity (the light intensity that produces the second-order correlation function) to the first-order correlation function that relates how the electromagnetic field interacts and is scattered by the illuminated sample structure. Standard DLS measurements cannot handle highly turbid samples (as gels often are), since they assume a single scattering event and are thus compatible only with dilute solutions. Moreover, systems undergoing irreversible dynamics (non-ergodic) require specialized setups where scattering is averaged by moving the sample and investigating different positions. On the other hand, diffuse-wave spectroscopy provides information about highly scattered samples, but in this case, a large number of scattering events is required to properly describe the light scattering as a diffusive process [53,54,55]. The intermediate range, that is, the dynamic one formed *during* the sol–gel transition, is poorly investigated since there are no analytical models able to handle the complexity of the transient between the sol and the gel regimes. In order to fill this gap and investigate the dynamics at the nanoscale of the sol–gel transition, we used the statistical properties of the light scattered during a DLS experiment and proved that they contained relevant information about the dynamics of the sol–gel transition, and from this analysis, we could distinguish samples that differed in the relative amounts of their components.

## 2. Results and Discussion

CNCs were structurally characterized with AFM, and a representative image of the CNCs is presented in Figure 1a. The results of the statistical analysis of CNC sizes is shown in Figure 1b. The nanocrystals had an average length of about 170 nm, broadly distributed with a St. Dev. of about 120 nm, while their height lay between 2 and 5 nm (averaged over more than 300 CNCs). DLS analysis confirmed these data, as reported in Figure 1c. The structure was compatible with the usual one obtained from the TEMPO-mediated oxidation synthetic pathway.

The presence of carboxylic groups on the surface of the CNCs was confirmed by FTIR spectroscopy and is reported in Figure 2; the most significant spectral region is below 1800 cm^−1^. The spectrum of pure and hybrid CNC gels (NA3 and NP3; see Table 1 for the sample composition) were vertically shifted and rescaled to make relevant spectral signatures overlap. No baseline corrections were applied. Looking at the CNC spectrum, the strong peak at about 1610 cm^−1^ marks the COO^−^ asymmetric stretching, while the one at 1420 cm^−1^ refers to the symmetric one. The C–O–C stretching is underlined by the peak at 1160 cm^−1^ [56,57]. The triplet between 1030 and 1110 cm^−1^ confirms the high crystallinity degree of the CNC [57]. Most intense peaks are tentatively assigned according to references above and [58,59,60,61]. Note that while the materials that created an extended network of intermolecular bonds (CNC and NA) show broad absorption peaks, the NP samples keep the detailed vibrational structure of the PEG alone, thus supporting the fact of the small affinity between ${\mathrm{Mg}}^{2+}$ cations and the polymer.

The spectra of the hybrid materials maintain the typical feature of the pure polymers, suggesting that the interactions that formed the gel did not significantly modify the vibrational degrees of freedom of the polymer chains. Consequently, the main role in forming the gel seemed to be that of the CNC, while CNC–polymer interaction was of less importance. This hypothesis is also supported by evidence that the stiffer gels were those formed by pure CNC (see Figure 3(left)).

Finally, the zeta potential of TEMPO-oxidized CNC is shown in Figure 1d and clearly shows a large negative value in the surface potential, compatible with those typically reported in the literature and mainly ascribed to the presence of deprotonated carboxylic groups. Rheological analysis characterized the elastic and viscous moduli ($G^{\prime }\;\mathrm{and}\;G^{\prime \prime }$) of the gels and the results within the LVER are reported in Figure 3(left). Generally, both moduli increase in the order CNC > NP > NA, and for both the CNC and NA, the elastic modulus increases proportionally with the concentration of CNC. This fact is compatible with the overall stiffening of the gel framework, despite the poor affinity of ${\mathrm{Mg}}^{2+}$ with alginates in the case of the NA hybrids. On the other hand, for a given CNC–PEG ratio, the NP samples show a nearly constant $G^{\prime }$ up to a ${\mathrm{MgCl}}_{2}$ concentration of 30 mM, and only above this value does their $G^{\prime }$ increase with a slope similar to that of the other compositions. Moreover, the NP samples have slightly larger *G* moduli for low ionic strength compared with alginate hybrids. These considerations are confirmed by the viscosity characterization reported in Figure 3(right). All hybrids showed shear thinning behavior typical of CNC suspensions [62], with a marked reduction in their viscosity over a range of about two orders of magnitudes at large shear frequencies. Pure CNC was the most viscous sample since the electrostatic interaction among CNCs was the strongest and the mesh size in the gel network was the smallest. All other samples showed similar viscosities (data for all samples investigated can be found in Appendix A).

It should be noted that, although the literature reports a variation in the elastic modulus over more than an order of magnitude (from $10^{3}\;\mathrm{to}\;10^{4}$ Pa [14,63]), the effectively exploitable range is limited by the equilibrium condition established within the gel during its actual use. Indeed, as already pointed out [15], the concentration of the crosslinking cation reaches a steady state value, independently of the amount of cations used during the gelation step. Moreover, increasing the mechanical properties of the gel by using a large excess of cations might produce unwanted results (e.g., toxic gels which release excess ions when used).

For applications such as cell cultures and, in particular, in the field of tissue engineering, where cells are embedded into an artificial ECM, it is of paramount importance to understand the actual state of the material internal architecture. DLS can provide information on the dynamics of gels during their aging and show how materials with similar macroscopic mechanical behavior might have very different microstructures and dynamics, which can impact their biocompatibility.

Figure 4 reports the dynamics of ($\sigma^{2}$) for representative samples for each investigated composition (data for all samples can be found in Appendix A). Initially, the systems were in a colloidal suspension state of barely interacting species and $\sigma^{2}$ tended to the ideal value of 2, typical of ergodic systems. Upon the addition of ${\mathrm{MgCl}}_{2}$, cations crosslinked the species into clusters of increasing size, and the suspension immediately jellified, creating a solidified blob within the cuvette. Since the ${\mathrm{MgCl}}_{2}$ solution was added at once within the DLS cuvette, the dynamics of cation diffusion were stochastic in nature and the induction time before $\sigma^{2}$ decreased could not be estimated in advance. Yet, all samples showed a sharp drop in $\sigma^{2}$ by roughly an order of magnitude after a variable induction time. The optimal threshold for each sample is indicated by the different background colors in Figure 4 panels. This phenomenon is the signature of a strongly reduced mobility in the particles (regardless of if they were CNCs or polymer molecules), which was induced by the large concentration gradient of cations that quickly rearranged the structure of the system, from a solution into a gel network. The fast dynamics within the scattering volume destroyed the coherence required to form significant autocorrelation; thus, $\sigma^{2}$ collapsed. Moreover, the $\sigma^{2}$ fluctuations testify that the initially formed gel network was a metastable structure that slowly evolved towards its equilibrium configuration. In the end, by tracking $\sigma^{2}$, it was possible to have a hint of an understanding of the dynamics of the sol–gel transition of the system.

The FOM for the five different compositions, for the optimal *T*-value, is reported in Figure 5.

The CNC and NP samples shared similar dynamics, with $\sigma^{2}$ decreasing sharply and mostly assuming small values after the transition occurred. On the other hand, NA samples showed a marked decrease in $\sigma^{2}$ from its initial value, but then it continued to oscillate for the entire time window considered. Pure CNC samples are reported for illustrative purposes and their FOM is comparable to those of the NP samples, suggesting similar gelation dynamics for the CNCs and NPs. The NAs had a markedly different behavior and, by increasing the amount of CNC in the hybrids, the FOM decreased, tending to the one of the pure CNC. This fact supports the idea that our phenomenological parameter effectively represents a physical feature of the internal dynamics of the gel.

The dynamics of $\sigma^{2}$ can be understood by the following reasoning: upon the addition of the crosslinker, ${\mathrm{MgCl}}_{2}$, a non-equilibrium jellifying blob formed nearly instantaneously. Then, strong diffusive fluxes established from the blob to equilibrate its composition. These diffusive processes destroyed the coherence required to measure the typical (flipped) sigmoidal profile of $g^{2}(t)$. Moreover, since these fluxes affected the entire volume of the cuvette, multiscattering events took place along the optical path of the laser, thus further decreasing the characteristic decay time and $\sigma^{2}$. The difference between CNCs/NPs and NAs was due to their different aptitude to jellify: while the CNCs and NPs formed “stable” structures upon coordination with ${\mathrm{Mg}}^{2+}$ cations, the NAs maintained their internal dynamics. Alginate chains acted as a viscous liquid reservoir that trapped jellified particles of CNC, and the latter were responsible for the fluctuations in the $\sigma^{2}$ function. Again, given the large concentration and the broad dispersion in the size of these CNC particles, no quantitative analysis could be performed of the $g^{2}(\tau )$ runs.

To verify the reliability of our approach, we repeated the analysis for the same sample compositions but used ${\mathrm{Ca}}^{2+}$; in these cases, the cation bound both PEG and alginates and the $\sigma^{2}$ always produced a small FOM, comparable to that of the pure CNC. This result confirms the strong affinity of ${\mathrm{Ca}}^{2+}$ with both cellulose and polymers and its role as an effective crosslinker.

Finally, the hybrids were used as substrates for cell cultures. HeLa cells were used as model cell lines and were seeded with an initial areal cell density of $3.5\times 10^{3}\;\frac{\mathrm{cells}}{{\mathrm{cm}}^{2}}$.

Cell cultures were incubated for up to 96 h. Cell viability was verified using standard MTT assays. The results are reported in Figure 6.

HeLa cells adhered within 24 h and exhibited growth over time on any of the five CNC-based hydrogels, with varying growth rates. Compared to the reference polystyrene dishes, all gels showed slowed cell growth, yet both CNCs and NPs showed a constant increase in the cell number, while NA hybrids initially hindered cell viability, even though they also showed an increase in the cell number with time.

It is well known that, apart from their chemical nature, the mechanical properties of substrates strongly affect processes such as cell–matrix interaction, cell signaling, and mechanotransduction [64,65]. Therefore, the viscoelastic properties of synthetic ECMs for cell culture must be carefully adjusted to promote cell growth.

Pure CNC hydrogels seem to be the most promising substrate for cell cultures. Notably, the viscoelastic properties of the hybrids were rather similar since their $G^{\prime }$ values differed by no more than a factor of three (for the 50 mM ${\mathrm{MgCl}}_{2}$ conc.) and yet they showed rather different cell growth rates. Data in the literature for such soft substrates demonstrate that cell viability is not affected by the actual value of their elastic moduli [66,67,68]; thus, given their similar viscoelastic properties, differences in cell spreading seem to be associated mainly with the microarchitecture of the gel network. Indeed, all three individual components (CNC, PEG, and alginate) are known to be viable substrates for cell growth. Thus, while CNCs and NPs have a similar stiffness, they differ for a fundamental functional group (the amino group), and they show similar growth rates; on the other hand, NAs and CNCs have very similar chemical compositions but different microarchitectures, and this is reflected in the largely different growth rates.

## 3. Conclusions

In this work, we present two results:We showed that both the composite hydrogels made of CNC/linear amino–PEG and CNC–alginate were good substrates for growth in a model cell line. However, CNC/amino–PEG induced much faster cell proliferation, despite the similar physical macroscopic properties of the two tested materials. This result suggests that the cells also responded to dynamic environmental cues not detectable by conventional techniques.We introduced a novel DLS analysis implemented with commercial, single-angle apparatus that was able to discriminate between gels with similar macroscopic mechanical properties but differing chemical composition and microscopic structural dynamics. As representative tests, we investigated the dynamics of CNC–PEG and CNC–alginate materials. To prove the relationship between the dynamic DLS data and the microscopic hydrogel structure, we exploited the peculiar characteristic of alginates jellifying in the presence of ${\mathrm{Ca}}^{2+}$ but not with ${\mathrm{Mg}}^{2+}$ [42], since smaller ${\mathrm{Mg}}^{2+}$, differently from ${\mathrm{Ca}}^{2+}$, does not fit into the G boxes and establishes with alginate chains a weaker affinity-driven interaction ruled basically by Manning’s theory [69]. By defining a phenomenological Figure of Merit (FOM), based on the short-time values of the second-order correlation function, we showed that its statistical properties contained information about the sol–gel transition and the material composition. We observed that the nanoscale dynamical properties were in some way retained when the hydrogels attained both stable and similar macroscopic properties. We suppose that the differences in their internal dynamics causally affected the hydrogel interaction with cells in the culture. In fact, while CNC–PEG composites formed a stable gel structure in a relatively short time and it induced faster cell proliferation, the slowly equilibrating CNC–alginate hydrogels showed longer induction time. Notably, cell proliferation depends on the stability of cell adhesion to the support and our results indicate that this stability is affected by nanoscale phenomena. Cell adhesion and proliferation are fundamental issues for the design of innovative hydrogels for human use (e.g., they are invaluable for tissue regeneration, wound healing, drug delivery, and organoids), and at present, nanoscale behavior is often neglected. Conversely, this behavior seems to play an important role, and here, we provide a simple approach for its monitoring. Thus, by combining the DLS approach to current state-of-the-art techniques (i.e., NMR, SAXS, and rheometry) we could obtain more exhaustive information on the gelation mechanism. In fact, DLS is characterized by a significantly better temporal (tens of μs) and spatial (tens of μm) resolution and it provides short-timescale dynamic information for small size clusters of a gel. The other advantage of the proposed approach is its easy implementation using conventional DLS equipment. While phenomenological in nature, our approach is the only one reported able to fully exploit the spatial and temporal resolution provided by DLS. It provides quantitative information about differences between materials that vary in specific parameters, like component ratios. Although DLS is a well-known technique, this is the first report where the statistical properties of the initial values of $g^{\left(2\right)}$(*t*) are used to derive information about the chemical composition and the dynamical state of a material. Our proof-of-principle demonstration will pave the way for a much broader use of our approach. Dedicated models and computational tools are needed to fully understand the method’s possibilities and limitations and will be developed in the future.

## 4. Materials and Methods

### 4.1. Materials

Commercial wood pulp, Celeste 85, was kindly provided by SCA (Sundsvall, Sweden). 2,2,6,6-tetramehylpiperidine-1-oxyl (TEMPO) 98+% was purchased from Alfa Aesar; NaBr 99% and NaOH 1 M produced by Carlo Erba (Milano, Italy) and NaClO EMPLURA solution with 6–14% active chlorine produced by Merck KGaA (Darmstadt, Germany) were used. Bis(3-aminopropyl)-terminated PEG, with average molecular weight of ∼1500 Da, purchased from Merck KGaA, and alginic acid sodium salt solution with very low viscosity, purchased from Thermo Fisher Scientific Chemicals (Waltham, MA, USA), were used to produce CNC hybrid hydrogels. All chemicals were used as received, without further purification. MilliQ-grade water was used for all preparations and processes. HeLa cells were cultivated in Dulbecco’s Modified Eagle Medium (DMEM) containing phenol red and supplemented with 10% Fetal Bovine Serum (FBS), 2 mM glutamine, 100 U/mL penicillin, and 100 μg/mL Streptomyosin (Pen-Strep). PBS (pH 7.4), Trypsin-EDTA 0.25%, and phenol red were used to wash cells before dissociation and for trypsinization, respectively. We used 0.4% Trypan Blue Solution to stain alive cells to count them in a Burker chamber hemocytometer (Merck KGaA (Darmstadt, Germany)). The viable cells were assessed on the basis of the mitochondrial activity by measuring the reduction of dimethyl thiazolyl diphenyl tetrazolium salt (MTT) to formazan crystals by mitochondrial dehydrogenases, which provides insights into cytotoxicity levels in cell cultures. For MTT assays, phenol red-free DMEM supplemented with FBS, Pen-Strep, and L-glutamine was used since it is proven that phenol red compromises the outcome of MTT assays. The CyQuanTTM MTT kit was used to assess cell viability. All the listed chemicals were purchased from Thermo Fisher Scientific Chemicals.

### 4.2. Synthesis of CNC

TEMPO-oxidized CNC was produced with a slightly modified receipt from [70]. Briefly, 10 g of cellulose pulp was swollen in 500 mL of distilled water for 5 h at room temperature under stirring and then sonicated in bath for 30 min. A total of 160 mg of TEMPO and 1 g of NaBr salt were added and the final volume was brought to 1 L. The oxidation was started by the addition of 35 mL of NaClO and left to proceed for 3 h, with the pH kept at 10.5 by the addition of 1 M NaOH. The slurry was rinsed 5 times with plenty of distilled water and divided in 35 mL aliquots. Each aliquot was sonicated with a Bandelin Sonopuls HD 2200 ultrasonic homogeneizer (Berlin, Germany) equipped with a 13 mm tip, settled at 160 W, and then vacuum-filtered over a 5 μm cellulose acetate filter (Sartorius (Göttingen, Germany)). The CNC suspension was heated at 70 °C in a water bath under magnetic stirring until a concentration of 7 mg/mL was obtained. Finally, the suspension was again vacuum-filtered to remove residual impurities.

### 4.3. Synthesis of Hybrid Gels

CNC–PEG hybrids (NP) were prepared in the following way: 200 mg of bis(3-aminopropyl)-terminated PEG was dissolved in 10 mL of deionized water and stirred until a homogeneous solution was obtained. Subsequently, HCl (1 M) was added until the pH reached 7.2 ÷ 7.4. The resulting suspension was then filtered using a 0.22 μm pore-size filter with a PES membrane. Following filtration, it was mixed with a 7 mg/mL CNC solution in two different volume ratios: CNC–PEG = 2:1 and CNC–PEG = 3:1. CNC–alginate hybrids (NA) were similarly prepared: 2.5 g of alginic acid sodium salt was dissolved in 100 mL of DI water to obtain a 2.5% *w*/*v* aqueous solution. It was then filtered with a 0.45 μm pore-size filter with PES membrane and later mixed with the CNC solution in two different volume ratios: CNC–alginate = 2:1 and CNC–alginate = 3:1. The several compositions of the hydrogels under study are summarized in Table 1. Hybrid hydrogels were crosslinked by dropping a 1 M ${\mathrm{MgCl}}_{2}$ solution with a pipette along the inner border of polystyrene (PS) cuvettes/multiwells already containing the CNC-based solution such that the final ${\mathrm{Mg}}^{2+}$ concentration was 47.6 mM (we will round it up to 50 mM for the rest of this paper).

**Table 1 gels-11-00197-t001:** Details of the composition of hydrogels. Numbers indicate the relative volumes of the constituents.

Name	CNC	PEG	Alginate
CNC	1	0	0
NP2	2	1	0
NP3	3	1	0
NA2	2	0	1
NA3	3	0	1

When used for cell culture experiments, all CNC-based solutions (pure CNC and hybrids) were sterilized in an autoclave at 1 bar and 121 °C for 1 h.

### 4.4. Hydrogel Characterization

CNC was structurally characterized via Atomic Force Microscopy. An NT-MDT SMENA head equipped with an NSG-10 tip (elastic const.: 11.8 N/m; typical resonant freq.: 240 kHz) was used to measure the average CNC dimensions.

Fourier Transform Infrared spectra of both pure CNC and hybrid materials were acquired with a Nicolet FTIR microscope (Thermo Fisher Scientific) at a standard spectral resolution (4 cm^−1^) and with a suitable number of acquisitions (typically 64) for a good signal-to-noise value.

The rheological properties of the hydrogels were evaluated with a Malvern Panalytic KINEXUS PRO+ rheometer (Malvern, United Kingdom) equipped with a plate–plate geometry (diameter: 20 mm). The upper geometry plate was lowered until it was in conformal contact with the top surface of the hydrogel, corresponding to a gap distance of 1 mm. Samples were preformed, directly transferred onto the bottom plate, and equilibrated with 5 min resting time in the instrument. Dynamic moduli and the elastic response under oscillatory stresses were assessed through strain–sweep experiments to estimate the LVER (frequency = 1 Hz; strain: 0.01% to 10%) and frequency sweep (frequency ranging from 0.1 Hz to 100 Hz, constant stress chosen within the LVER). The share rate dependence of viscosity was studied within a shear ramp of γ = 1–100 s^−1^. All measurements were repeated at least three times and carried out at a constant temperature of 25 °C. Dynamic light scattering (DLS) characterization was carried out with an Anton-Paar Litesizer 500 DLS (Graz, Austria) machine, used with the 90° angle configuration. In this case, 100 μL of 1 M ${\mathrm{MgCl}}_{2}$ aqueous solution was added directly into the DLS cuvette that contained 2 mL of any of the CNC suspension. The DLS measure started soon after the addition of the cations and lasted for hours. The carboxyl content of CNCs was tested by conductometric titration and was found to be 1.6 μmoles/mg of cellulose.

### 4.5. Cell Culture and Cell Viability

Hela cells were used, since this cell line is considered a versatile model in testing the biocompatibility of substrates [71]. The cells were directly seeded on the material under study and the survival and growth rate were quantified by colorimetric, fluorimetric, and cytometric assays.

To assess cell survival on the hydrogels, two 48-wells plates were used to culture HeLa cells, extracted from the same cell line, to perform MTT assays after 48 and 96 h of incubation. Triplicates of the five CNC-based hydrogels under study were prepared in each multi-well plate by adding 20 μL of sterile 1 M ${\mathrm{MgCl}}_{2}$ aqueous solution to each well containing 400 μL of CNC-based solution for each composition. Prior to use, the five CNC-based suspensions were autoclaved, while the ${\mathrm{MgCl}}_{2}$ aqueous solution was filtered with a 0.22 μm-pore-size filter. The multi-well plates containing the hydrogels were left unperturbed overnight to complete gelation and then they were rinsed 5 times with sterile DI water before 2 mL of pure phenol-red DMEM was added to each well. In this way, DMEM could diffuse inside the hydrogels and exchange with water under the effect of the concentration gradient. After at least 3 h of incubation at 37 °C and in a 5% *v*/*v* CO_2_ atmosphere, the excess supernatant above each hydrogel was removed and substituted with 2 mL of freshly supplemented phenol red-free DMEM to increase the amount of nutrients stored inside the gel. After a second incubation of at least 3 h, another exchange with supplemented phenol red-free DMEM was performed, such that the concentrations of FBS, Pen-Strep, and L-glutamine inside the gels could increase up to the standard ones. HeLa cells were seeded on the hydrogels with an areal cell density of 10^3^ cells/mL, which was spotted as the optimal one during a preliminary trial. The same areal cell density was seeded in triplicate on pure polystyrene as a reference. MTT assays were performed at two different incubation times (48 h and 96 h) to verify cells’ interaction with the substrates and their growth rate. After incubation, the MTT protocol was followed to quantify the metabolic activity of HeLa cells: 1 mL of sterile PBS, thermalized at 37 °C, was added to one vial containing 5 mg of MTT to prepare the 12 mM MTT solution to be dropped in each well with a cell culture. Indeed, after having substituted the old culture medium with 300 μL of fresh, supplemented DMEM without phenol red in each well, 30 μL of the MTT solution was added to each well for testing. After 4 h of incubation, 300 μL of 0.1 mg/mL SDS solution at a pH of 2 was added to the culture medium to dissolve the formazan crystals formed by alive cells. After 15 h of incubation, the absorbance of the culture medium in each well was read at 570 nm with an Infinite M200 PRO microplate reader from Tecan (Männedorf, Switzerland). For the reading, the area of each well was sampled 25 times.

### 4.6. DLS Analysis

According to the objective of our research, we investigated the possibility of using DLS data to unravel the nanoscale dynamics of the 3D hydrogel network formation during an ionotropic gelation process. Below, a brief survey of the relevant equations is reported to clarify the idea behind the DLS data analysis performed, while more detailed descriptions can be found in the dedicated literature cited above.

In DLS, an electromagnetic field is used to track the dynamics of scatterers. If there is relative movement between the scatterers and the illuminated volume, then the scattered light fluctuates over time. The detector of the DLS tracks these fluctuations and constructs the so-called time-average normalized intensity correlation function: $g^{\left(2\right)}(k,\tau )=\frac{{\langle I(k,0)I(k,\tau )\rangle }_{T}}{{\langle I(k,0)\rangle }^{2}}$, where $k=\frac{4\pi n}{\lambda }sin(\theta /2)$ is the k-vector of the scattered light, *n* is the solvent refractive index, λ is the laser wavelength (elastic scattering is assumed), and θ is the angle at which light is collected with respect to the incident laser direction. $<\ldots {>}_{T}$ indicates a time average, that is, each measurement (run) is divided into *N* temporal bins and the numerator of $g^{\left(2\right)}\left(\tau \right)$ is $\frac{1}{N}\sum_{i=1}^{N}\chi \left(i\cdot T\right)\cdot \chi \left(i\cdot T+\tau \right)$, where χ is the number of photons collected within the iT-th bin (the normalization value of the denominator is of no relevance for our purposes). This analysis was performed by a dedicated electronic (the correlator) [46].

In the case of dilute colloidal solution, where scatterers freely diffuse with Brownian dynamics, long-enough measures allow the average over all possible configurations of the colloids to be obtained, that is, $g^{\left(2\right)}\left(\tau \right)$ contains all the information to fully describe the average system dynamics. On the contrary, in systems where scatterers are not free to move (e.g., within a gel), the temporally averaged $g^{\left(2\right)}\left(\tau \right)$ does not sample all the possible structural configurations of the scatterers and it provides a biased (i.e., incomplete) estimation of the system dynamics. These systems are described as non-ergodic and the proper sampling of their properties requires ensemble averaging that is a time-average replicated in several different spatial positions to be sure to collect the dynamics from most of the possible structural configurations and, thus, to obtain a reasonable estimate of the system dynamics.

The statistical behavior of $g^{\left(2\right)}\left(\tau \to 0\right)$ changes depending on if the system is (non-)ergodic. In particular [47], the following is observed:$g^{\left(2\right)}\left(\tau \to 0\right)=2$ in solutions (ergodic systems).$1<g^{\left(2\right)}\left(\tau \to 0\right)<2$ in gels.$g^{\left(2\right)}\left(\tau \right)∼1$ in glasses (completely frozen systems).

Generally, DLS retrieves information about the particle size by assuming dilute and ergodic samples. In this case, the field scattering time-correlation function is given by the Siegert relation: $g^{\left(2\right)}\left(\tau \right)={\left|g^{\left(1\right)}\left(\tau \right)\right|}^{2}+1$, where $g^{\left(1\right)}\left(\tau \right)$ is the Laplace transform of the characteristic decay rates $g^{\left(1\right)}\left(\tau \right)=\int_{0}^{\infty }G\left(\Gamma \right)\cdot exp\left(-D\Gamma \tau \right)d\tau$, where *D* is the characteristic decay time (i.e., the time the colloid takes to flow across the illuminated volume, which depends on both the solvent’s viscosity and the colloid size). In the case of monodispersed samples, there is just a single *D* and $g^{\left(1\right)}\left(k,\tau \right)=exp\left(-Dk^{2}\tau \right)$. By exploiting the Stokes–Einstein equation, the hydrodynamic radius of the colloid is extracted: $R_{H}=\frac{k_{B}T}{6\pi \eta D}$.

The gel building blocks are not free to diffuse. A basic model describes the gel scatterers as fluctuating around an average stable position. The overall light scattering is the superposition of two contributions:One coming from the “frozen”, averaged configurations of the scatterers;One due to the fluctuations around their average position.

Despite this apparently simple decomposition of the motion, the formulation of a model able to quantitatively describe the gel case is a formidable task yet to be solved and it requires detailed knowledge of the actual system under study (both physical and chemical details).

Actual DLS correlators routinely record delays down to hundreds of ns. These timescales are much faster than the typical transit time of a colloidal particle over the volume illuminated by the laser. In fact, consider a spherical colloid of about 100 nm in size, freely diffusing in water at 298 K. From the Stokes–Einstein equation, D $∼10^{-12}$ m^2^/s. For Brownian motion in 1D systems, the average displacement is $<x^{2}>=2Dt$. Considering a 20 μm-diameter illuminated volume, the time the colloid takes to cross it is $t=\frac{<x^{2}>}{2D}∼10\;s$. This demonstrates the idea that the first points of $g^{\left(2\right)}\left(\tau \to 0\right)$ contain information about the fast gel dynamics and can be used to monitor its structural evolution as well as its internal dynamics. To simplify the notation, in the rest of this paper, we define $\sigma^{2}=g^{\left(2\right)}\left(k,\tau \to 0\right)$, since this is the relevant value to define our FOM. It should be remarked that the information provided by our approach is complementary to and non-overlapping with that provided by rheology. In fact, while DLS investigates the microscopic system’s dynamic evolution, rheology monitors the macroscopic gel’s slow-timescale evolution, even at the sol–gel transition point.

DLS measurements were performed as follows: each gel composition was measured in triplicate, and for each measurement, we recorded several hundreds of autocorrelation runs, each of 5 s in duration. The exact number of runs depended on the sample since the induction time before $\sigma^{2}$ decreased could not be controlled (it depended on how ${\mathrm{MgCl}}_{2}$ diffused in the cuvette). The FOM was defined as the frequency of the fluctuations in $\sigma^{2}$ and it was calculated as follows: We averaged the first 20 points of each run to calculate the value of $\sigma^{2}$ (these points corresponded to the time interval between 200 ns and 1 μs). Then, we defined a threshold value (see below) for the $\sigma^{2}$ function (called *T*) and we located the time when $\sigma^{2}$ decreased below this threshold (at time $t=t_{0}$). To normalize the frequency vs. the duration of the experiment, we calculated the frequency at which $\sigma^{2}$ crossed *T* as the number of crossings (*c*) divided by the duration of the measurement $t_{T}<t<t_{f}$: $FOM=c/(t_{f}-t_{T})$, with $t_{T}$ is the time corresponding to the threshold and $t_{f}$ the total duration of the experiment. The optimal threshold was selected using a dynamic programming algorithm [72]; it aimed at minimizing the cost function with respect to the optimal partitioning of the dataset:(1)$$V\left(T,y\right):=\sum_{k=0}^{K}c\left(y_{t_{k}}^{t_{k+1}}\right)$$
where *V* is the quantitative criterion function to be optimized (minimized in our case), and $T=\{t_{1},t_{2},\ldots t_{k}\}$ is the set of indexes that partition the signal *y*. Thus, *V* is given by the sum of the cost functions (denoted by *c*) for each signal partition $y_{t_{k}}^{t_{k+1}}$. Since we know the number of thresholds in our data (a single threshold), the optimal solution is computed by(2)$$\min_{|T|=K}V\left(T,y\right)=\min_{0=t_{0}<t_{1}<\ldots t_{K}<t_{K+1}=T}\sum_{k=0}^{K}c\left(y_{t_{k}}^{t_{k+1}}\right)$$

## Figures and Tables

**Figure 1 gels-11-00197-f001:**
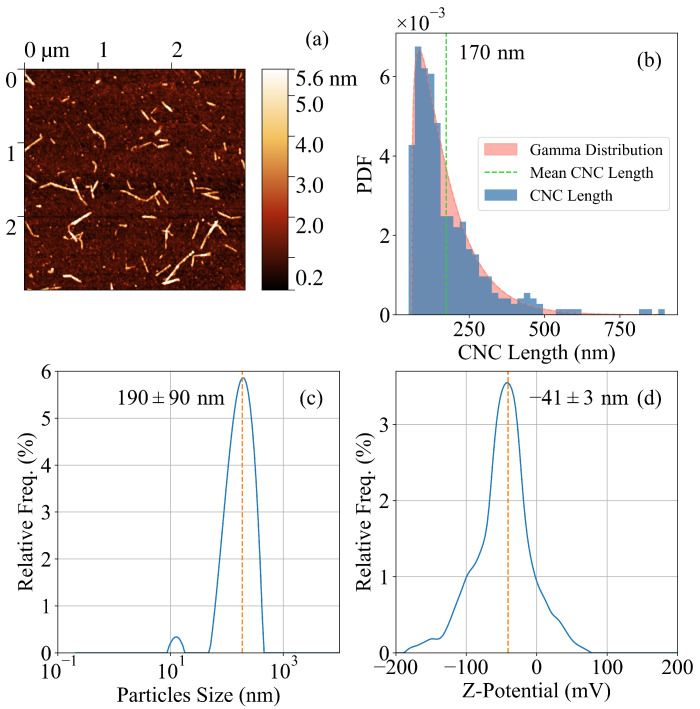
(**a**) A tapping-mode AFM image of a representative CNC. (**b**) A histogram of the probability distribution function of CNC size. (**c**) The DLS analysis of a similar CNC sample confirms the average size of the nanocrystals. (**d**) The Z-potential analysis shows the negative surface potential of CNCs due to the presence of hydroxyls as well as carboxylic groups.

**Figure 2 gels-11-00197-f002:**
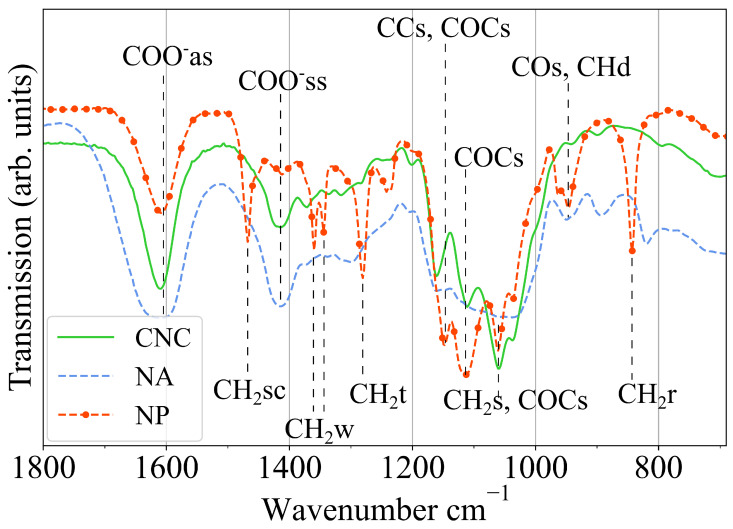
FTIR spectra of CNC (green continuous line), CNC-alginate (blue dotted line), and CNC-PEG (red dot-dash line) hybrids. Main peaks are labeled according to following: as: asymmetric stretching; ss: symmetric stretching; s: stretching; sc: scissoring; w: wagging; t: twisting; r: rocking; d: deformation.

**Figure 3 gels-11-00197-f003:**
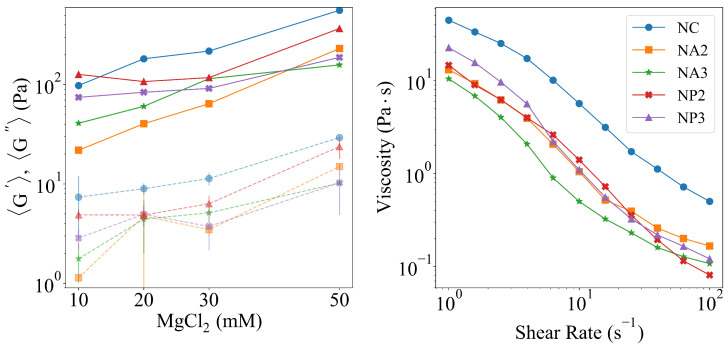
(**left**) $G^{\prime }$ (solid lines) and $G^{\prime \prime }$ (dashed lines) obtained from strain–sweep experiments at increased concentration of ${\mathrm{MgCl}}_{2}$. (**right**) Viscometry shear rate ramp.

**Figure 4 gels-11-00197-f004:**
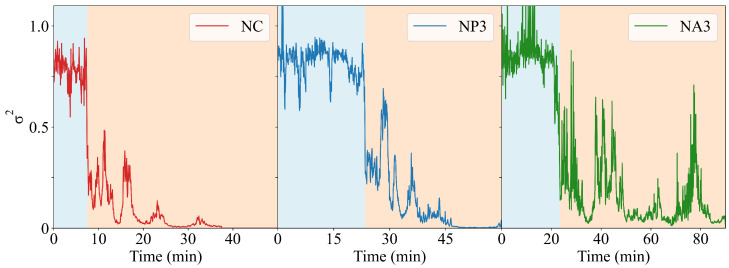
$\sigma^{2}$ dynamics versus time for a representative sample of each composition (indicated by the legend). The different background colors indicate the optimal threshold. Data for each sample investigated are reported in the ESI.

**Figure 5 gels-11-00197-f005:**
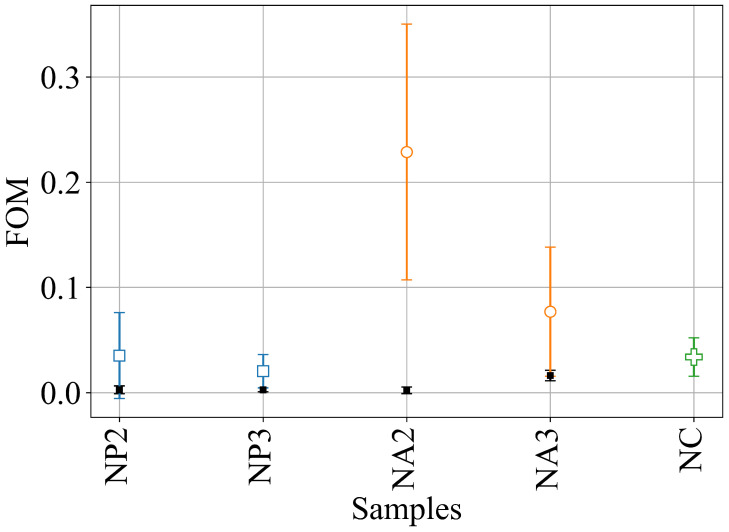
The classification of the different samples based on the FOM (using an L1 norm cost function). Coloured dots refer to the gelation using Mg as crosslinker; small black dots are for same hybrid compositions but those that were jellified using ${\mathrm{Ca}}^{2+}$.

**Figure 6 gels-11-00197-f006:**
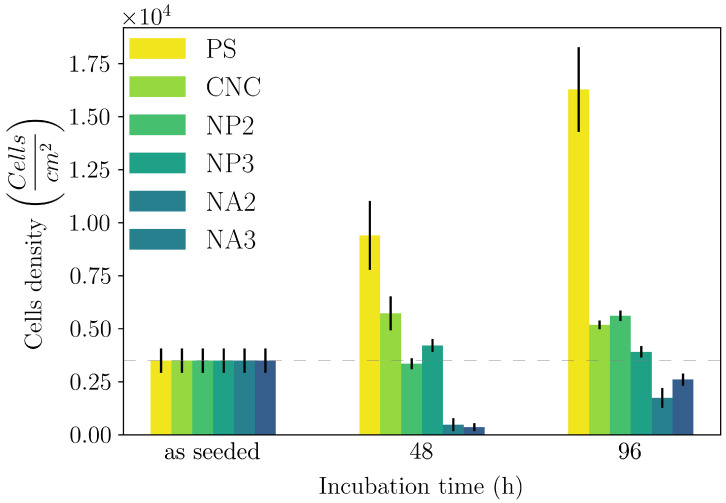
The results of the MTT assays: metabolic cell viability expressed as areal cell density vs. incubation time for the reference polystyrene and hydrogels. The dotted horizontal line is a guide for the eye, corresponding to the seeded areal cell density.

## Data Availability

The raw DLS and rheometry data used in this article are available at 1 February 2025 https://drive.google.com/file/d/1-OyLK4-ADz1NnPX9Op_dAlTNcxl5Ifkw/view?usp=sharing and https://drive.google.com/file/d/1OItGVoETq4vu82Uk8fkViyL1MEV0BI2U/view?usp=sharing.

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
