# Peer review of "Ion-Specific Gelation and Internal Dynamics of Nanocellulose Biocompatible Hybrid Hydrogels: Insights from Fluctuation Analysis"

_gels, 2025, doi:10.3390/gels11030197_

Round 1

Reviewer 1 Report

Comments and Suggestions for Authors The text is mostly clear and concise and is suitable for publication after minor changes as noted.   The study reports the use of Mg²⁺ as a crosslinker, but no discussion is provided on the reproducibility of gelation kinetics across different batches. Given that ionotropic gelation can be sensitive to ionic strength variations, was any control experiment performed to assess batch-to-batch consistency in the FOM values?   The TEMPO-oxidized CNCs used in the study have negatively charged surface groups. Given that the zeta potential of CNCs strongly influences their aggregation and crosslinking, how do the authors ensure that variations in the oxidation process (e.g., degree of carboxylation) do not affect the gelation behavior? Was the carboxyl content quantified?   Have the authors conducted extended swelling/deswelling cycle experiments to assess potential structural degradation, CNC leaching, or PEG diffusion over time? This would be crucial for evaluating the hydrogel suitability for long-term applications.

Reviewer 2 Report

Comments and Suggestions for Authors

The paper titled "Ion-Specific Gelation and Internal Dynamics of Nanocellulose Biocompatible Hybrid Hydrogels: Insights from Fluctuation Analysis" explores the behavior of hybrid hydrogels composed of nanocellulose combined with either amino-polyethylene glycol or sodium alginate. The authors report that their amino-containing hybrid gel exhibits biocompatibility, as demonstrated using a model cell line. They also suggest that fluctuations in the second-order correlation function from DLS experiments can provide valuable insights into the internal dynamics of these materials throughout the sol-gel transition, as well as quantitative information about ion-specific gel formation. This work integrates experimental data with a novel theoretical framework developed by the authors.

To prepare cellulose nanocrystals (CNC), they used conventional TEMPO oxidation on wood pulp. The resulting CNC was then converted into CNC-PEG hybrid gels and mixed with an alginate solution. Upon adding an Mg²⁺ solution as a crosslinker, changes in rheological properties and light scattering behavior were monitored. The study also establishes a correlation between Mg²⁺ concentration and shear rate.

Conceptually, the work can be divided into two main parts. The first part focuses on the kinetics of gel formation, while the second investigates cell growth on the formed gels. The authors introduce a phenomenological Figure of Merit (FOM), derived from the short-time values of the second-order correlation function, to compare two families of materials: Ca²⁺-crosslinked gels and Mg²⁺ suspensions. However, the biological part appears somewhat disconnected, as it does not directly build on the theoretical framework established in the first part. Moreover, the materials tested do not show a significant advantage over the polystyrene standard. After 96 hours, cell density remains at a similar level—or even slightly lower in some cases.

However, the manuscript is well-written and quite well-structured, with clear figures and properly cited references. This work somehow integrates experimental data with a novel theoretical framework developed by the authors. I recommend it for publication with the following minor comments:

  1. My main concern is the authors' claim that DLS experiments can replace rheometry, SAXS/SANS, and NMR techniques. This statement is debatable and requires at least a clarification of the criteria used for comparison. A more rigorous approach would involve an additional experimental section where SAXS/SANS and NMR data for this specific system are analyzed and compared with DLS results using these criteria.
  2. It is unclear whether a higher FOM value is better or not. Please clarify.
  3. Line 413 – MTT assay: Please provide a brief explanation of what this entails.
  4. Lines 440–441 – You mention that “the cells respond to dynamic environmental cues not detectable by conventional techniques.” Could you elaborate on the mechanism behind this response and suggest techniques that might be able to detect it?
  5. Figure 1: In the AFM image, it would be beneficial to include a cross-section profile of the sample. Outline in which surface you applied your gel? The roughness of the surface plays essential role!
  6. Figure 2 – It would be better to label the peaks directly in the figure rather than in the supplementary material.
  7. Figure 6 caption – What do “PS” and “NC” refer to? If PS stands for polystyrene, consider defining this acronym somewhere in the text.
  8. For all figures, I see LaTeX formatting. While this is fine, consider using Times New Roman or Arial for better readability.

Reviewer 3 Report

Comments and Suggestions for Authors

This work presents the preparation and characterization of four CNC-based hybrid hydrogels blended with amino-PEG or alginate. Overall, this manuscript presents an interesting study, particularly the use of DLS to develop a method for characterizing internal gel dynamics. However, before it can be considered for publication, several issues must be addressed.

  1. Line 325, the peak is not at 1600 cm-1, it should be at a higher wavenumber. Please revise.
  2. In Figure 3, comparing to pure NC, why adding PEG first increase the modulus and then decrease the modulus (NC>NP2>NP3)? Is there any interaction between PEG and NC that leads to this? Please clarify.
  3. Line 406, the reviewer believes that the Ca2+ experiment is a crucial part of the study, but the relevant data is not shown.
  4. Figure 6, why does alginate substantially decreases the cell viability? Please comment.
Comments on the Quality of English Language

Some grammatical errors:

Line 5: 'a simplified receipt'

Line 45: 'an hybrid gel'

Line 53: 'Amines functionalities'

Please proofread

Reviewer 4 Report

Comments and Suggestions for Authors

Ref.:  gels-3502042

The manuscript entitled “Ion-Specific Gelation and Internal Dynamics of Nanocellulose Biocompatible Hybrid Hydrogels: Insights from Fluctuation Analysis”

In this study, this article presents biocompatible nanocellulose-based hydrogel's synthesized without intermediate steps. It uses Dynamic Light Scattering to reveal internal dynamics and ion-specific gel formation with higher resolution than conventional methods. Although the study is well-executed and written, I suggest the manuscript be considered for publication in the Gels journal, provided the authors address the following minor revisions.

  1. It would be beneficial if the authors could highlight the unique aspects of their work in the introduction as a separate paragraph.
  2. How do internal dynamics influence long-term cell proliferation, adhesion, and viability in extended experiments?
  3. The authors should include preliminary results from the computational study to strengthen the manuscript.
  4. In Table 2, the authors' rationale for not using a 2:2 ratio requires further explanation.
  5. In Figure 2, assigning specific vibrational bands with their specific functional group in the FTIR spectrum would make it easier for readers to understand.
  6. The author should incorporate the standard deviation for Figure 3, to provide a clearer understanding of the data.
  7. All abbreviations should be spelt out when first mentioned in the manuscript. For example, PEG, CNC, DMEM, and PEGDA should be defined in the introduction.
  8. Ref [35] should be cited consistently in the introduction, following the same format as other references throughout the manuscript.
  9. In the Materials and Methods section, "TEMPO" appears twice on line 130; it should be mentioned only once.
  10. At line 174, there should be no gap between ◦ C.
  11. The author should check all the references such as adding the DOI for references 38, 39, 40, 47, 49, 51, and 58 respectively.

Round 2

Reviewer 4 Report

Comments and Suggestions for Authors

Dear Authors,

I want to inform you that all of my comments have been properly addressed.